# Nucleoprotein as a Promising Antigen for Broadly Protective Influenza Vaccines

**DOI:** 10.3390/vaccines11121747

**Published:** 2023-11-23

**Authors:** Alexandra Rak, Irina Isakova-Sivak, Larisa Rudenko

**Affiliations:** Department of Virology, Institute of Experimental Medicine, St. Petersburg 197022, Russia; alexandrak.bio@gmail.com (A.R.); isakova.sivak@iemspb.ru (I.I.-S.)

**Keywords:** influenza virus, nucleoprotein, influenza vaccine, cross-protection, conserved protein

## Abstract

Annual vaccination is considered as the main preventive strategy against seasonal influenza. Due to the highly variable nature of major viral antigens, such as hemagglutinin (HA) and neuraminidase (NA), influenza vaccine strains should be regularly updated to antigenically match the circulating viruses. The influenza virus nucleoprotein (NP) is much more conserved than HA and NA, and thus seems to be a promising target for the design of improved influenza vaccines with broad cross-reactivity against antigenically diverse influenza viruses. Traditional subunit or recombinant protein influenza vaccines do not contain the NP antigen, whereas live-attenuated influenza vaccines (LAIVs) express the viral NP within infected cells, thus inducing strong NP-specific antibodies and T-cell responses. Many strategies have been explored to design broadly protective NP-based vaccines, mostly targeted at the T-cell mode of immunity. Although the NP is highly conserved, it still undergoes slow evolutionary changes due to selective immune pressure, meaning that the particular NP antigen selected for vaccine design may have a significant impact on the overall immunogenicity and efficacy of the vaccine candidate. In this review, we summarize existing data on the conservation of the influenza A viral nucleoprotein and review the results of preclinical and clinical trials of NP-targeting influenza vaccine prototypes, focusing on the ability of NP-specific immune responses to protect against diverse influenza viruses.

## 1. Introduction

Respiratory viral infections are frequent human pathologic conditions caused predominantly by seasonal viruses [1]. Although the course of these infections is usually mild, there are dangerous complications associated with opportunistic diseases that can exacerbate the course of comorbid conditions, especially in the elderly, young children, and immunocompromised individuals [2]. One of the most dangerous respiratory viral infections is influenza, which poses a serious threat to the global economy and public health and requires the development of effective means of prevention and treatment, particularly taking into account the variability of pathogens [3].

The most effective measure of prevention against influenza is vaccination, which provides individual protection and serves as the basis for herd immunity [4]. At present, there exist several types of licensed influenza vaccines, including inactivated, live-attenuated, and recombinant protein vaccines. In addition, various alternative vaccine platforms, such as mRNA, DNA, viral vectors, and recombinant nanoparticles, have been investigated [5]. Viral surface proteins are most commonly used as targets for vaccine development as they mediate the interaction of the virus with the target cell receptors, the fusion of the viral and cell envelopes, and the entry of the pathogen into the host cell. Indeed, the abundance of these molecules on the virion surface, along with their usually high immunogenicity and active expression within infected cells, enables their effective presentation to the immune system, as manifested by pronounced humoral and T-cell responses during subsequent infection. However, this property of viral surface proteins also makes them a major target for viral evolution toward host defense evasion and increases their susceptibility to escape mutations [6]. This variability of surface antigens requires an almost annual update of influenza vaccine strain compositions and often leads to reduced vaccine effectiveness when there is an antigenic mismatch between circulating and vaccine strains [7,8]. Therefore, the development of influenza vaccines with broader and more durable protection against seasonal and potentially pandemic influenza strains is of high priority [9]. Various strategies have been explored to develop more broadly protective influenza vaccines, where highly conserved viral antigens may be used as a target for the induction of immune responses [10,11]. In particular, one of the most common conserved viral antigens used for universal influenza vaccine design is nucleoprotein (NP), a highly abundant internal protein involved in the transcription and replication of the viral genome, influencing the host specificity and virulence of viruses [12,13]. As NP sequences are conserved not only between different strains within the same subtype but also between different influenza A virus subtypes or between two influenza B virus lineages, their use as a target antigen has become one of the main strategies for the development of cross-protective influenza virus vaccines [14,15,16].

It has been generally recognized that the key mechanism of the immune system response to this antigen is the activation of the T-cell mode of immunity (CD8+ cytotoxic T-lymphocytes) through presentation by infected cells of NP-derived peptides in complex with major histocompatibility complex-I (MHC-I) molecules, followed by their killing [17]. In addition, an infected organism typically produces a significant amount of anti-NP antibodies with high affinity that do not exhibit neutralizing activity but are associated with a reduction in disease severity [18]. Although this antigen is not exposed on the surface of virions, it has been detected on the surface of virus-infected cells [19,20], making it a potential target for antibody-mediated immune responses, such as antibody-dependent cellular cytotoxicity (ADCC) or complement activation [21,22].

Although NP is considered to be a highly conserved protein of influenza viruses, there is a slow accumulation of mutations in this antigen as a result of genetic evolution and positive selection by host immunity [23]. In this review, we summarize current knowledge on the conservation of influenza A viral nucleoprotein and review the results of preclinical and clinical trials considering NP-targeted influenza vaccine prototypes, focusing on the ability of NP-specific immune responses to protect against diverse influenza viruses.

## 2. Nucleoprotein Structure and Functions

In general, the organization of the NP–genome complex and domain structure of the NP molecule are similar to those of the majority of ssRNA(-) respiratory viruses. The NP protomer includes two folded domains: an N-terminal domain (NTD) and a C-terminal core domain (CTD). It is widely accepted that the site of nucleic acid binding is usually located between the NTD and CTD lobes, forming a clamp for the viral antigenome [24] (Figure 1). The influenza virus NP molecule of influenza virus binds to 24 nucleotides, and this number is fixed [25,26]. As the NP protein tends to bind the host RNA in the absence of viral RNA, its protomers are chaperoned by phosphoprotein to maintain the monomeric state and prevent the formation of empty or defective virions [27].

The crystal structures of recombinant NP–RNA complexes have been described as regularized formations [28,29], resulting from a spontaneous assembly of multimeric forms by the interactions of the corresponding domains [14,30]. In the infected cell, NP occurs in a number of forms: its monomers are complexed with a phosphoprotein; NP oligomers are bound with RNA and organized into ribonucleocapsid, and multimeric NP constitutes the relaxed structure making the RNA accessible for polymerase [31].

The multiple functions of the NP protein in the viral life cycle include the following: (i) protection of viral RNA from degradation by cellular enzymes (by physical RNA separation and masking from immune recognition, (ii) fitting the helical structure of RNP, (iii) modulation of the transcription and replication of viral RNA templates in a histone-like manner, and (iv) induction of immunosuppression (inhibition of effector cytokine synthesis and FcR-medicated signaling) upon infection [32].

NP was initially not considered a priority target for influenza vaccine development, as this antigen is not exposed on the surface of the viral particle, and anti-NP antibodies do not possess neutralizing activity. However, further studies have indicated that a large number of virus-specific T cells are produced to NP epitopes and that anti-NP antibodies could mediate cellular protective reactions. Since the 1980s, this antigen has become an attractive target for the development of universal influenza vaccines, mainly due to the high conservation rate of the NP among antigenically distant viruses.

**Figure 1 vaccines-11-01747-f001:**
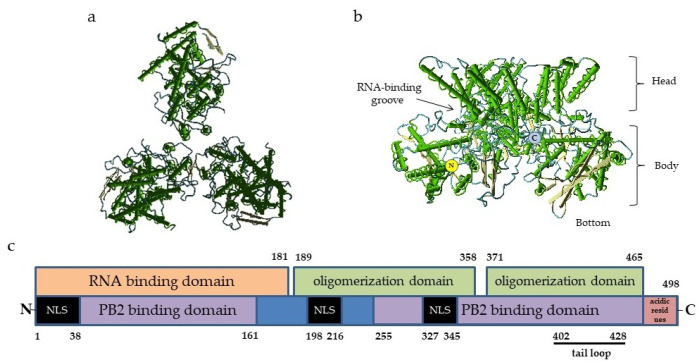
Structure of influenza virus NP: (**a**)—Top view of nucleoprotein trimer along the nucleocapsid axis, PDB ID 2IQH. (**b**)—Side view of a nucleoprotein trimer; the structures of the molecule are indicated, PDB ID 2IQH; and (**c**)—the scheme of NP molecule domains involved in the binding of RNA and other proteins. The numbers refer to the positions of amino acid residues. The C-terminal acidic residues (indicated in red) serve as repressors of PB2 and NP binding. The black bar indicates the tail loop region responsible for the oligomerization. NLS, nuclear localization signal. The locations of the NP functional domains are given according to [28,31,33,34].

In general, the NP antigen is known to mainly elicit both cytotoxic CD4+ and CD8+ T-cell (CTL) responses, as NP-derived peptides may be synthesized by the infected cell or produced as a result of exogenous RNP phagocytosis, which then may be presented in complex with both MHC class I or II, respectively [35]. These two cell populations are able to distinguish virus-infected cells from intact cells and similarly perform specific cytolysis by granules containing perforin and granzymes [36], thus playing a pivotal role in pathogen elimination. Therefore, targeted vaccination inducing T-cell responses against conserved NP epitopes is considered to be a promising strategy for the establishment of cross-protective antiviral immunity.

Unlike another internal influenza antigen, PB1, which is equally targeted by CD4+ and CD8+ T cells [37], NP is believed to be a major target for CD8+ CTLs [17,38]. Nevertheless, CD4+ CTLs constitute a minor population of CD4+ T cells (mainly represented by T helpers) [39] and have been described as primarily targeting NP and M antigens upon infection with the influenza virus [40,41]. This fact underpins the feasibility of the development of new vaccination strategies for the elderly using vaccines based on CD4+ CTL NP epitopes, as it is well known that with age, the number of CD4+ CTLs increases, but the activity of these cells upon influenza vaccination remains unchanged [36].

As NP has been detected at the membranes of infected epitheliocytes [19], it may be targeted by the humoral immune mode. Unfortunately, little is known about the patterns of generation, mechanism of action, and protective potential of anti-NP antibodies, which are actively produced in response to infection or vaccination [21,42]. In theory, they may contribute to viral clearance, mediating innate killing reactions, such as ADCC or antibody-dependent cellular endocytosis [43]. This hypothesis is supported by data on their ability to activate natural killer (NK) cells ex vivo [21] and by the killing of influenza-infected cells in vitro induced by sera from individuals with intensive anti-NP humoral responses but low anti-HA and anti-NA titers [22]. The function of anti-NP antibodies is most likely mediated by FcεRI receptors [44].

Notably, although the major components of the immune system involved in the anti-NP response are known, additional studies are needed to clarify the details that should be taken into account when developing effective epitope-based universal influenza vaccines.

## 3. Development of NP-Based Broadly Protective Vaccine Prototypes

The nucleoprotein of the influenza virus has been considered an attractive target for the design of broadly protective influenza vaccines for almost four decades [45], due to the T-cell responses elicited with respect to this antigen after infection, as well as after immunization with whole-virion vaccines, being highly cross-reactive. Indeed, an early study found that cytotoxic T cells produced in mice primed with the influenza virus of a particular subtype could recognize transfectant mouse L cells expressing a NP of a different subtype, whereas no recognition was observed in heterologous HA-expressing L cells [46].

The first attempt to use nucleoprotein as a vaccine antigen was reported by Wraith et al. in 1985. In their study, a modified X31 (H3N2) influenza virus was treated with bromelain four times to release the majority of the surface glycoproteins [47]. The resulting core was further treated with ammonium deoxycholate and centrifuged at 10,000× *g* for 30 min. The supernatant was highly enriched with NP, although residual HA protein was also present. This NP/HA preparation was injected i.p. (50 µg) into BALB/c mice, and 3 weeks later, high levels of X31 virus-specific cytotoxic memory T cells were detected in mouse splenocytes, suggesting that NP was a major inducer of T-cell immunity [47]. To further test this hypothesis, the same group of investigators prepared more pure NP protein extract (containing less than 3% HA impurity) from the X31 virus and used it for immunogenicity and cross-protection studies in mice [48]. The main finding of this study was that 75% of the NP-primed mice survived a lethal challenge with the heterologous H1N1 virus, although the clinical manifestation of the disease was not significantly improved. These data suggest that the induced NP-specific cytotoxic T cells contributed to viral clearance but did not prevent the infection [48].

A further attempt to improve the NP-based vaccine was made by Tite et al., who expressed NP from the A/NT/60/68 (H3N2) influenza virus in *Salmonella typhimurium* and studied this protein as a vaccine antigen in mice. Similar to the studies with purified NPs, i.p. immunized mice were protected against lethality caused by a heterologous H1N1 virus, but still displayed considerable morbidity before recovery. Importantly, when these mice were boosted i.n. with the same antigen, improved protection was observed including accelerated clearance of the virus from the lungs. Furthermore, these i.n. boosted mice also rapidly cleared an antigenically unrelated influenza B virus [49].

A key mechanism of NP-mediated protection is thought to involve the triggering of T-cell immune responses. To make this possible, the antigen should be produced inside the target cell on the matrix of the exogenous mRNA/DNA delivered by the viral vector or lipid nanoparticles. mRNA platforms seem to be a promising tool for the development of NP-based vaccines due to their low risk of insertional mutagenesis, high potency, accelerated development cycles, and cost-effectiveness [50]. Some vaccine candidates comprising mRNA-expressing NP coated with a lipid layer have been proposed [18,51,52]; however, the best results regarding the stimulation of T-cell responses and protection against viral challenges were achieved when NP mRNA was used in combination with mRNAs encoding other influenza antigens.

Several attempts have been made to create vaccines based on NP-encoding DNA, which was intended to be administered in an adjuvant-free manner via the i.m. route. In their early study, Ulmer et al. demonstrated that an NP-based DNA vaccine evoked T-cell responses mediated by antigenic presentation through MHC-I and provided cross-protection against lethal influenza viruses in a murine challenge model [53]. Furthermore, the authors demonstrated the lymphoproliferative and Th1-type cytokine-secreting effects of the NP DNA injection. In particular, increases in the levels of IFNγ and IL-2 mediated by CD4+ T cells as a result of restimulation of splenocytes with the NP antigen in vitro were shown. T-cell depletion followed by a viral challenge revealed the effector action of CD4+ and CD8+ lymphocytes generated in response to immunization with NP DNA [54]. Epstein et al. found that mice primed with NP DNA and boosted with NP encoded by the adenoviral vector showed higher titers of specific serum IgG, as well as a significant increase in the number of effector T cells, and were better protected against the lethal challenge with various influenza variants—including the highly pathogenic H5N1 virus—than mice vaccinated with NP DNA alone [55]. Furthermore, Lo et al. used DNA and adenoviral vectors expressing NP, M2, or NP+M2 antigens to demonstrate that “DNA prime-adenoviral vector boost” vaccination induces systemic humoral and cellular responses, as well as protection against a homologous challenge in the same manner as immunization with cold-adapted viruses. However, only vaccination with DNA and the adenoviral vector encoding both NP and M2 antigens could confer protection against a challenge with the H5N1 virus [56]. Rao et al. tested the efficacy of a similar prime-boost vaccination regimen in mouse and ferret models. According to their results, HA-, NP-, and M2-bearing DNA- or adenovirus-based vaccines were able to elicit specific humoral responses, but only vaccination with HA-encoding or mixed DNAs and adenoviral vectors fully protected the animals from a lethal dose of a pathogenic H5N1 virus [57].

As a tool for delivering the NP gene into cells, several vector platforms have been proposed. In this line, a recombinant vector MVA-NP+M1 bearing the influenza antigens NP and M1 and propagated in chicken embryo fibroblasts or AGE1.CR.pIX cells was developed by Berthoud et al. [58]. To date, this vaccine has passed phases I and IIa of the clinical trials and shown an ability to stimulate a rapid T-cell response across different age groups and to boost pre-existing levels of specific T cells in aged individuals [59]. However, there were no statistically significant improvements in the outcomes of the disease caused by i.n. infection of vaccinated subjects [60]. Furthermore, to boost pre-existing cross-reactive T cells capable of protection against heterosubtypic influenza A viruses, MVA-NP+M1 was combined with a licensed inactivated influenza vaccine and tested in people over 65 years of age in phase IIb [61,62]. Despite an increase in the magnitude of the T-cell responses, the study could not demonstrate an increase in hemagglutination-inhibiting antibody titers or an association of disease outcomes with the number of IFNγ+ T cells. A pentavalent NP-encoding vaccine developed on the backbone of the Wyeth strain of vaccinia (a vector similar to MVA) presented immunogenicity and protective potential in an H5N1 challenge study [63].

Later, Dicks et al. constructed a novel replication-deficient safe vector for the expression of vaccine antigens based on the chimpanzee adenovirus ChAdOx1 [64]. The vaccine candidate ChAdOx1-NP+M1 turned out to be safe when administered to volunteers and was able to stimulate T-cell immune responses [65]. A similar vector was used by Arunkumar et al., who demonstrated the efficacy of priming with ChAdOx1-NP+M1 and boosting with MVA-NP+M1 for inducing the vigorous expansion of IFNγ+ CD4+ T cells. At the same time, the best protection against a challenge with H3N2, H7N9, and H10N8 viruses was achieved in the case of vaccination with the vectors simultaneously expressing chimeric HA, NP, and M1 [66]. Further study in a ferret model confirmed the efficacy of immunization with these ChAdOx1- and MVA-based triple-expressing vectors in a prime-boost regimen to abrogate replication of the H3N2 virus in the respiratory tract [67]. The effects of prime-boost immunization of pigs with ChAdOx2- and MVA-vectored vaccines expressing NP, M1, and NA antigens depended on the administration route: i.m. injection evoked systemic antibody responses, while aerosol immunization led to mucosal formation of specific T cells and IgGs. Notably, shedding of the H3N2 virus challenge and lung damage were reduced in both cases [68].

Adenovirus-associated virus (AAV)-vectored vaccines expressing H1, M1, and NP have been constructed by Sipo et al., which were shown to evoke potent specific T-cell and serum IgG formation in mice. However, only immunization with a trivalent vaccine improved the survival of mice challenged with the virus belonging to the same subtype [69]. In contrast, the results of Demminger et al. suggested that immunization of mice with AAV-based vaccines expressing NA, HA, or cHA alone allows not only for the induction of pronounced systemic and mucosal humoral responses, but also protects the animals against a homologous challenge [70]. The study of Lee et al. described the protective properties of the NPs of two influenza B lineages encoded in the replication-deficient Ad5 vector. The authors revealed the induction of cross-recognizing antibodies and CD8+ T-cell responses, as well as cross-protection against heterologous viruses in a mouse challenge model [16]. To develop a tool for the control of influenza in swine, the parapox orf virus was used as a vector for the expression of NP and/or HA of a swine influenza virus. The resultant vaccine candidates expressing HA and NP or HA alone were tested in pigs and were found to elicit cross-neutralizing antibodies accompanied by robust virus-specific T-cell responses. Interestingly, immunization with the vector containing both HA and NP genes led to the induction of a larger number of CD8+ T cells, the development of a Th2-biased immune response, and incremental protection against challenge infection [71].

In theory, the immunogenicity of recombinant NP may be improved with the addition of costimulatory adjuvants or by fusion with functional exogenic or oligomerization domains. Zheng et al. described the testing of a universal influenza vaccine based on the recombinant NP protein expressed in a bacterial system and administered to mice three times in combination with or without the C48/80 adjuvant. Although high serum IgG titers were detected in all of the NP-immunized mice, the challenge with different influenza strains at a lethal dose revealed the best protection in the case of i.n. administration of NP in complex with C48/80 [72]. Similar results have been obtained by Guo et al., who revealed the ability of an i.n. administered vaccine, based on the combination of recombinant NP of the A/PR/8/34 (H1N1) influenza virus and cholera toxin, to provide complete protection against the homologous virus challenge and partial protection against the heterologous H5N1 and H9N2 avian influenza viruses [73]. Another promising adjuvant for the recombinant NP protein, bis-(3′,5′)-cyclic dimeric adenosine monophosphate (c-di-AMP), has been proposed by Sanchez et al. The formulation was administered i.n. and was found to stimulate NP-specific serum and mucosal antibodies, as well as promoting robust Th1-type responses accompanied by increases in INFγ and IL-2 levels. The protective efficacy of the vaccine against weight loss and lung tissue damage was also described [74]. In a study by Cookenham et al., recombinant NP was administered to young and aged mice in complex with a novel squalene-based adjuvant SLA-SE. The study revealed a modest increase in the numbers of NP-specific CD4+ and CD8+ T cells in splenocytes and intensive generation of anti-NP IgG2c antibodies, as well as viral clearance intensification and increased survival rates; however, the authors found no significant changes in the number of memory CD8+ T cells in vaccinated mice [75]. Li et al. reported that the combination of NP and M1 recombinant influenza proteins provoked cytotoxic anti-NP (but not anti-M1) T-cell responses and conferred protection against a homologous virus challenge only when mice were immunized with proteins supplemented with a physical radiofrequency adjuvant (RFA). However, the RFA-adjuvanted antigens did not increase the levels of anti-NP and anti-M1 antibodies, compared to proteins mixed with the AddaVax adjuvant, and did not increase the serum level of the proinflammatory cytokine IL-6. Thus, RFA likely acts as a mild T-cell response stimulator [76]. Furthermore, the authors managed to enhance NP-induced CD8+ T-cell responses and improve the protection against lethal viral challenge infections by using the CpG 1018 adjuvant, instead of the AddaVax. This change in adjuvant also led to a switching of the type of immune response to a Th1-biased (major antiviral effector pathway mediated by IFNγ-secreting cells) instead of a Th1/Th2-balanced response (mediated by IL4-producing T lymphocytes) [77].

The induction of high levels of serum and mucosal anti-NP antibodies and a robust T-cell response have been observed after the i.n. immunization of mice with recombinant NP fused with a cell-penetrating trans-activator (TAT) protein from the human immunodeficiency virus-1 (HIV-1). This easy-to-deliver vaccine variant provided effective protection against the PR8, H9N2, and H3N2 viruses in a murine challenge study [78]. Using the same strategy, Tan et al. created a fusion protein consisting of recombinant NP as a displaying component and M2e antigen, both to study the changes in antigenicity of these proteins caused by point mutations and to propose a universal influenza vaccine candidate. While NP substitutions affected the binding of this antigen with the anti-NP antibody, mutated M2e continued to be recognized by anti-M2e IgG [79].

As another tool for more effective delivery and antigenic presentation of recombinant NP, the i.n. administration of the chitosan-based nanoparticles containing an inner NP(H9N2) core and outer HA1(H9N2) layer has been described. These stable nanobodies were generated using the ionic gelation method. Higher titers of anti-HA and anti-NP antibodies and an increased number of activated CD4+/CD8+ T cells were found in chickens immunized with HA1/NP nanoparticles compared with inactivated vaccine administration, which resulted in reduced viral shedding upon challenge [80]. Other novel double-layered nanoparticles, composed of HA- or M2-enriched shells and an NP-containing core that were able to induce neutralizing anti-HA and anti-M2 antibodies and ADCC activity, have been proposed by Ma et al. as a new candidate for influenza prophylaxis. In addition, immunization with these conglomerates stimulated robust NP-specific T-cell responses. As expected, the best protection of mice against virus challenges was achieved when using a trivalent vaccine combining HA, NA, and NP antigens, as evidenced by a histopathological analysis of the lungs and survival dynamics [81]. In another study, Wei et al. developed nanoparticles consisting of recombinant NP encapsulated by an apoferritin (AFt) nanocage conjugated with HA molecules exposed on the outer surface of the AFt. This vaccine prototype induced cross-reactive anti-HA and NP-specific antibodies in mice and provided complete protection against a homologous PR8 H1N1 and a heterologous A/FM/1/47 (H1N1) virus challenge, whereas the variant lacking NP conferred only 60% protection against the heterologous strain. Such a conjugation strategy may be potentially used not only for the creation of nonadjuvanted vaccines, but also as a tool for targeted drug delivery [82].

An attempt to improve the immunogenicity of recombinant NP from the A/WSN/1933 (H1N1) virus by antigenic multimerization has been made by Del Campo et al., who supplemented the NP sequence with the oligomerization domain OVX313 and developed the NP heptamer named OVX836, which consists of seven copies of the NP antigen [83]. This polymer appeared to be significantly more immunogenic—due to the more active uptake by dendritic cells, compared to wild-type NP—and stimulated robust NP-specific CD4+ and CD8+ T-cell responses, particularly lung-localized CD8+ tissue-resident memory (T_RM_) cells in mice [84]. In a phase 1 clinical trial, OVX836 administered intramuscularly was found to be safe and immunogenic at a dose of up to 180 µg [85]. In a recent phase IIa clinical trial, higher doses of OVX836 (300 or 480 μg) were compared to the 180 μg dose in terms of safety, immunogenicity, and protection against seasonal influenza-like illnesses in a particular influenza season. All vaccine doses demonstrated a good safety profile and the induction of pronounced cellular responses in a dose-dependent manner, which was evidenced by increases in the numbers of IFNγ-secreting cells in peripheral blood as early as a week after immunization [86]. Importantly, this study demonstrated an approximately 80% efficacy against seasonal H3N2 influenza viruses, indicating the ability of the induced T cells to protect vaccinated individuals from natural influenza infection. However, this study did not investigate the role of anti-NP antibodies, and this mode of immunity could also contribute to protection against infection.

Table 1 summarizes the most significant findings from the preclinical studies of NP-based universal influenza vaccine prototypes based on various vaccine platforms.

NP-based vaccines are not commercially available, but many of the prototypes have been assessed in clinical trials (Table 2). In addition to the candidates described above, the results of clinical trials considering several other vaccine prototypes have been published. For example, the vaccine Multimeric-001, which is a recombinant protein comprising the conserved linear epitopes of HA, NP, and M1 antigens mixed with Montanide ISA-51 oil adjuvant, was shown to be safe and immunogenic in phases I and II of the clinical trials [97]. In addition, Multimeric-001 has been studied in the elderly as a priming tool prior to administration of the trivalent inactivated influenza vaccine; in this age group, such a vaccination regimen provided intensified antiviral antibody generation, compared to unprimed individuals [98,99]. Despite these encouraging results, this vaccine did not facilitate a reduction in influenza severity and thus failed to induce specified outcomes in a phase III clinical trial in older adults (NCT03450915). A possible cause of vaccine inefficiency could be weak epitope-specific T-cell formation, which was insufficient for protection against the disease.

The same oil-based compound, Montanide ISA-51, has been used as an adjuvant in a synthetic influenza vaccine (FLU-v) containing the peptides of the NP, M1, and M2 viral proteins. Being s.c. administered, this vaccine induced vigorous virus-specific CTL responses in healthy participants in a phase I clinical trial [107]. A positive correlation of this effect with a milder disease course and reduced viral loads was observed in a following human challenge with a wild-type H3N2 influenza virus [106]. The immunogenicity of FLU-v in healthy volunteers has been further revealed in a phase 2b clinical trial [111]. A H1N1 human challenge study [109], as well as a phase 3 trial study [110], confirmed its efficacy for influenza prevention.

Little is known about the clinical efficacy of the FP-01.1 vaccine, consisting of six fluorocarbon-modified 35-mer peptides comprising the conserved epitopes of NP, M1, PB1, and PB2, which are able to trigger CD4+ and CD8+ T cells. This candidate was proposed to prime and boost vaccination against different influenza A variants and has demonstrated safety and the ability to stimulate cross-reactive IFNγ-secreting CD4+ and CD8+ T cells at a dose of 150 μg/peptide in a phase I trial [113]. The results of further clinical trials on this vaccine have not yet been published (Table 2).

## 4. Variability of NP Sequences and Implications for the Performance of NP-Based Vaccines

Although it is widely accepted that the NP protein is highly conserved among different subtypes of the influenza A virus, this antigen undergoes slow evolutionary changes as a result of the virus escaping recognition by the CTLs [113]. Several previous studies have demonstrated that a number of critical amino acid substitutions occur in the NP protein and that these mutations are associated with escape from human CTLs [114,115,116,117]. It has been shown that the substitutions in epitopes including the 103–111, 251–259, 380–388, 383–391, and 418–426 amino acid residues—in particular, mutations at positions 103 (K203R), 259 (S259L), 384 (R384K/G), 421 (D421E), 423 (P423S), and 425 (I425M)—may abrogate the antigenic presentation mediated by MHC-I and, thus, allow the virus to avoid recognition by CD8+ CTLs [116,118]. Such changes may lead to a complete inability of effector T cells generated in response to infection with previous influenza variants to recognize the viruses belonging to recent strains [118].

A possible consequence of these mutations in T-cell epitopes is that CTLs targeted at nonmutated epitopes will inefficiently recognize cells infected with the mutated virus. To demonstrate the variability in the NP sequence, we aligned the amino acid sequences of NPs that are most often used as vaccine antigens, along with recent seasonal H1N1 and H3N2 influenza viruses, specifically, A/WSN/1933 (H1N1) and A/PR8/34 (H1N1) (Figure 2 and Appendix A). We also included NP sequences of master donor viruses of a licensed live-attenuated influenza vaccine (LAIV), as they are known to induce robust NP-specific T-cell responses [119,120] (Appendix A). As shown in Figure 2, a number of mutations had occurred in the NP sequences of more recent viruses, suggesting that T cells targeted to some NP epitopes of the developed NP-based vaccine candidates, as well as to that of the licensed LAIVs, may not recognize the corresponding proteins of currently circulating influenza A viruses. For example, an immunodominant NP_418–426_ epitope has up to four mutated residues in recent H3N2 viruses, compared to the original model viruses. Although CTLs specific to this particular epitope are characterized by high functional avidity [121], it is very unlikely that T cells produced in response to vaccination with the nonmutated NP will recognize NP_418–426_ epitopes with three or four mismatched residues.

Earlier, we performed a simple immune epitope conservancy analysis of the NPs of currently circulating H1N1 and H3N2 viruses, relative to the NPs originating from the master donor virus of a licensed LAIV [123]. The analysis revealed that 8 out of 18 (44.4%) A/Leningrad/134/17/57 (H2N2)-derived NP epitopes had been conserved among recent H1N1 and H3N2 viruses, suggesting that most of the predicted NP-specific immunodominant CTL epitopes of the vaccine prototypes developed from older influenza A viruses are no longer present in the vast majority of circulating influenza viruses. These findings may have at least two negative consequences: (i) the massive implementation of the NP-based vaccines designed on the backbone of obsolete influenza viruses will result in the stimulation of CTL clones that will fail to recognize cells infected with recent seasonal influenza viruses, thereby unnecessarily overloading the immune system of vaccinees; and (ii) many of the new NP epitopes that have emerged in recent influenza viruses will not be recognized by vaccine-induced T cells, as these epitopes were not yet present in the older vaccine viruses.

It is well known that, in addition to the T-cell response, the NP antigen can induce the production of NP-specific antibodies. A number of studies describing the protective properties not only of recombinant NP but also of passively transferred immune sera from NP-immunized animals have demonstrated the efficacy of such vaccination in a heterosubtypic influenza virus challenge. In a study by Carragher et al., immunization of mice with recombinant NP resulted in high titers of specific IgG and reduced replication of the virus challenge, but not in antibody-deficient animals. Similarly, passive immunization of naive mice devoid of B cells or intact mice with anti-NP serum significantly reduced virus titers in the lungs [124]. LaMere et al. have shown that anti-NP antibodies administered to mice with depleted B cells provided partial heterologous protection and a reduced viral load through mechanisms mediated by FcRs and CD8+ T cells. Notably, FcR-deprived mice immunized with anti-NP antibodies appeared to be less-protected, suggesting an FcR-mediated effect of anti-NP antibodies in vivo [44]. The high immunogenicity of the NP protein and the ability of anti-NP monoclonal antibodies of influenza patients to mediate Fc-associated protective responses have been demonstrated by Varderven et al. [125]. In contrast, passive immunization of mice with human anti-NP mAbs was insufficient to protect against the virus challenge. These studies suggest that effective antiviral protection in vivo may be provided not only by anti-NP antibodies, but also by other influenza-specific defense factors [125]. In the study of Fujimoto et al., transgenic mice expressing anti-NP-antibody genes from B cells of H5N1 convalescents were exposed to a lethal dose of this highly pathogenic avian virus. The resulting antibodies had no neutralizing activity, but provided resistance to the animals against infection by both H5N1 and mouse-adapted H1N1. These data suggest that anti-NP antibodies are involved in cross-protection against influenza, probably through indirect antiviral reactions, such as ADCC or complement cascade activation [126]. Overall, NP-specific antibodies may contribute significantly to cross-protection induced by NP-based vaccines, such that this mode of immunity should not be disregarded. Although NP-specific antibodies are considered highly cross-reactive, the use of anti-NP monoclonal antibodies revealed some variations in the antigenicity of this protein between evolutionarily diverse influenza viruses [127]. In addition, several mAbs have been described that can only bind to a specific subtype of influenza A virus, suggesting a critical influence of some NP residues on antibody binding [128]. In the study of Bodewes et al., human monoclonal anti-NP antibody bound diverse H1N1 and H3N2 viruses with different intensities, indicating variations in the affinity of the antibody for the respective NPs [19]. All of these data suggest that slow evolutionary changes in the NP protein may also affect its binding to anti-NP antibodies, which should also be considered during the development of broadly reactive NP-based influenza vaccines.

## 5. Approaches to Improve the Performance of Influenza Virus Vaccines

There are several ways to reduce NP epitope mismatches between the vaccine antigen and currently circulating seasonal influenza viruses. First, for vaccines based on recombinant proteins, viral vectors, and nucleic acid platforms, more recent NP antigens should be considered in the development of vaccine prototypes. There are no clear obstacles preventing the replacement of the NP antigen in the experimental vaccine candidates with that of a recent influenza A virus, other than conducting additional experiments to confirm its stability and safety. However, NPs of seasonal influenza viruses of H1N1 and H3N2 differ quite significantly; therefore, further studies may be needed to assess the rational design of a universal NP antigen that includes all of the most important immunodominant B- and T-cell epitopes of both virus subtypes. For example, consensus sequences of influenza virus hemagglutinin molecules have been successfully generated using a COBRA approach (computationally optimized broadly reactive antigens), and these consensus HA vaccines demonstrated broad protection against heterologous influenza viruses within a particular subtype [129,130,131]. As the NP protein is inherently much more conserved than the HA antigen, the use of COBRA algorithms is likely to generate some NP consensus sequences that would induce broadly protective antibody and cell-mediated immunity. Furthermore, the performance of NP-based vaccines can be significantly improved by using new potent adjuvants and delivery platforms (as reviewed in [132,133,134]).

For live-attenuated influenza vaccines, the situation is somewhat different. Currently licensed LAIVs are based on the cold-adapted backbone viruses A/Leningrad/134/17/57 (Len/17) and A/Ann Arbor/6/60 (AA), which were isolated in 1957 and 1960, respectively [135,136]. These viruses were serially passaged at low temperatures and acquired unique mutations in the internal protein genes, which made the viruses temperature-sensitive and cold-adapted in vitro and attenuated in vivo. These mutations have been clearly identified in both master donor viruses [137,138] and, although no common mutations were found in the AA and Len/17 viruses, all substitutions affected the viral RNA synthesis machinery [139]. Therefore, a reasonable idea arose to develop vaccine strains against various human and animal influenza A viruses by simply introducing these attenuating mutations into the genes of the virus polymerase complex [140,141,142]. A side-by-side comparison of the Len/17 and AA mutations introduced into the A/PR8/34 backbone virus demonstrated that the Len/17 mutations resulted in higher attenuation of the virus compared with the AA mutations, although the immunogenic and cross-protective properties were comparable [143]. However, for some wild-type viruses, introduction of the full set of *ts* signatures from the cold-adapted AA virus was not enough to render the virus sufficiently attenuated; this depended mainly on the source of the PB2 gene [144]. Importantly, a recent study has found that the addition of a mutation L319Q in the PB1 protein significantly improved the attenuation of the H1N1pdm-based LAIV candidate carrying AA-specific attenuating mutations [145].

It should be noted that the strategy of introducing attenuating mutations directly into the genome of wild-type viruses results in LAIVs with the most relevant repertoire of B- and T-cell epitopes across the viral proteome, and such vaccines should obviously provide the highest possible level of protection against circulating influenza viruses [146]. However, the procedure for preparing such vaccine candidates is laborious and time-consuming, as vaccine strains against both H1N1 and H3N2 seasonal influenza viruses must be prepared through side-directed mutagenesis. Therefore, it is unlikely that LAIVs against seasonal influenza viruses will be routinely prepared in this way. Moreover, there will always be the possibility of the emergence of wild-type virus variants whose pathogenicity cannot be compensated for by a standard set of attenuating mutations, as was the case with the 2009 H1N1 pandemic variant [140].

In the meantime, there is a simple and straightforward way to improve the immunogenicity and cross-protection of currently licensed LAIVs by simply replacing the NP gene of the cold-adapted master donor virus with that of a wild-type influenza virus, such as switching from a 6:2 genome composition to a 5:3 composition [147]. In this case, there is no need to design a “universal” NP protein that includes the most important immunodominant B- and T-cell epitopes of both virus subtypes, as traditional LAIVs contain both the H1N1 and H3N2 vaccine strains. Given that nucleoprotein is a major target antigen for influenza-specific cross-reactive cytotoxic T cells [17,148], updating epitopes even in this protein alone is expected to lead to more broadly protective influenza vaccine variants. Of note, the NP gene of the Len/17 strain does not contain attenuating mutations and can be replaced by the wild-type NP without safety concerns [137]. Indeed, a panel of H3N2 5:3 LAIV candidates developed on the Len/17 backbone were equally safe and immunogenic in a ferret model, when compared to classical H3N2 6:2 LAIVs, confirming the feasibility of switching to 5:3 LAIV genome compositions [149]. It should be noted that the differences in NP-specific T-cell responses between 5:3 and 6:2 LAIVs were not tested in ferrets; however, a series of experiments on C57BL/6J mice demonstrated that NP_366_ epitope-specific CTL responses differed significantly between the 6:2 and 5:3 LAIVs of the H7N9 subtype. In particular, the 6:2 LAIV induced a robust CTL response (both in splenocytes and in lungs) to the homologous Len/17 NP_366_ epitope, whereas very low reactivity was observed for the H7N9 NP_366_ epitope [123,150], suggesting that classical LAIV induces an excessive level of T-cell responses to outdated NP epitopes, and these T cells unnecessarily overwhelm the immune system as they no longer recognize evolutionarily diverse viruses. Overall, animal experiments have demonstrated that the Len/17-based 5:3 LAIVs of different subtypes afforded better protection against a heterologous viral challenge than standard 6:2 LAIVs, prompting their further evaluation in clinical trials [123,150,151,152]. Importantly, experiments on PBMCs isolated from HLA-typed blood donors have confirmed that 5:3 LAIVs could stimulate human influenza CD8+ T cells more relevant to current infections than classical 6:2 LAIVs [153].

For LAIVs based on the A/Ann Arbor/6/60 backbone, switching to the 5:3 genome composition may be complicated as one of the *ts* mutations was mapped to the NP protein (D34G) [138]. However, recent H1N1 viruses have naturally acquired this mutation (Figure 1); therefore, at least for this subtype, 5:3 LAIVs may be developed without safety concerns. However, further studies are needed to confirm this assumption.

## 6. Concluding Remarks

It is generally recognized that T cells have a major contribution to immunity against influenza and other respiratory pathogens by helping to eliminate pathogens from an infected individual [154,155]. This type of immunity is typically more broadly reactive and lasts longer than antibody responses to major viral antigenic determinants. Therefore, it is not surprising that the influenza virus nucleoprotein has attracted much attention as a target antigen for the development of broadly protective influenza vaccines, as a high proportion of T cells elicited by the infection recognize NP-specific epitopes. Clearly, NP is one of the most widely used antigens for designing universal influenza vaccines, along with the ectodomain of matrix 2 protein (M2e) [156].

To date, many experimental NP-based influenza vaccines aimed at inducing a cross-reactive T-cell responses, with the overall goal of protection against multiple influenza A subtypes, have been submitted to clinical or preclinical testing. These vaccines have been developed using different vaccine platforms, such as recombinant proteins fused to different adjuvants or oligomerizing domains, DNA/mRNA vaccines, or viral vector-based vaccines. Almost all published studies have demonstrated a broader spectrum of action of experimental vaccines; however, not all vaccines that have reached clinical trials have shown efficacy in humans. One possible reason for the insufficient degree of protection against current wild viruses is the mismatch between the epitope composition of the NP protein in the vaccine virus and the current circulating strain(s), as most vaccines are developed on the basis of well-characterized NP proteins from viruses isolated almost 90 years ago. During this period, despite a fairly high degree of NP conservation, a large number of mutations have arisen, some of which can be categorized as immunogenic T-cell epitopes. Furthermore, it is well known that even a single amino acid mismatch in a T-cell epitope can dramatically alter the immunogenicity of a peptide [157].

Although live-attenuated influenza vaccines stand apart from broad-spectrum NP-based vaccines, it is well known that they induce a potent T-cell response, and a major target for this response is also a nucleoprotein. Backbone viruses for LAIV were isolated more than 60 years ago, and there is also a serious mismatch between the NP epitopes of vaccine virus and that of wild-type strains. Preliminary findings in LAIVs expressing NPs of recent influenza viruses suggest that the source of the NP antigen significantly affects the magnitude of the T-cell response to the NP epitopes of current influenza viruses; therefore, the 6:2 LAIV genome composition should be switched to the 5:3 formula, with the NP derived from wild-type viruses, along with HA and NA.

In addition to cell-mediated immunity, the NP antigen generates high levels of anti-NP antibodies. This mode of immunity has been undeservedly ignored in preclinical studies of NP-based influenza vaccine prototypes, perhaps because anti-NP antibodies do not possess a neutralizing activity. However, these antibodies can bind with virus-infected cells and mediate protection through some Fc-dependent functions [22,124,158]. It is reasonable to suggest that slow evolutionary changes in the NP can also affect the antigenicity of the protein and lead to the reduced binding of anti-NP antibodies raised to immunization with the developed NP-based vaccines with the NP antigen on the cells infected with contemporary influenza viruses. Similar to influenza NP, the SARS-CoV-2 nucleocapsid (N) protein slowly evolves, and although the N-based serologic tests developed on the ancestral strain are still sensitive enough to detect antibody responses to evolved SARS-CoV-2 variants [159], these mutations may lead to substantial changes in the immunogenicity and antigenicity of the NP protein [160].

In conclusion, this review summarized recent advances in the design of broad-spectrum vaccines employing the nucleoprotein protein as the vaccine antigen. We showed that the particular antigen choice for vaccine design can have important implications for vaccine efficacy against current circulating influenza viruses and that even minor epitope mismatches between the vaccine and circulating viruses can significantly reduce vaccine performance. Finally, it is essential to conduct side-by-side analyses of some vaccine prototypes with different sources of NP proteins to determine more precisely whether it is really necessary to use NPs from more recent influenza viruses to design NP-based vaccines, or whether the cross-reactivity of vaccines based on older proteins yields a comparable performance to vaccine candidates with updated NPs.

## Figures and Tables

**Figure 2 vaccines-11-01747-f002:**
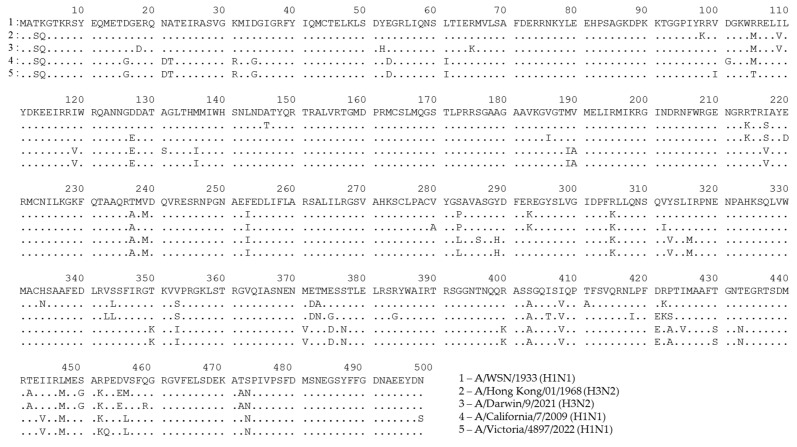
Alignment of NP sequences for A/WSN/1933 (H1N1) influenza virus and recent H1N1 and H3N2 seasonal influenza strains. This figure was generated using GISAID data and the uGene v33.0 software [122].

**Table 1 vaccines-11-01747-t001:** Examples of NP-based influenza vaccines.

Carrier and Antigen	Animals	Immunization Route	Main Results	Ref.
Purified NP of X31 (H3N2)	Mice	i.p., i.m., or s.c.	Mice were immunized i.p. (50 µg) or i.m. (10 µg) or s.c. (10 µg) with NP, once or twice. Vaccination induced cytotoxic T-cell responses toward the H3N2 and H1N1 viruses. Immunized mice were protected against lethality after challenge with the PR8 virus, although significant body weight loss was observed.	[48]
Recombinant NP of A/NT/60/68 (H3N2)	Mice	i.p. or i.n.	Mice were immunized twice i.p. with 10 µg of rNP precipitated with alhydrogel. Four weeks after second dose, some mice were i.n. immunized with 5 µg of NP in PBS. The i.p./i.n. prime-boost resulted in better protection against the lethal PR8 virus than i.p. immunization only. This regimen also resulted in better clearance of influenza B virus from the lungs. rNP did not induce CTLs, and the protection was mediated by CD4+ T cells.	[49]
DNA encoding NP of PR8 virus	Mice	i.m.	A total of 100 µg of NP DNA was injected into each leg at 0, 3, and 6 weeks. NP-specific IgG antibodies were detected already after the first immunization. The vaccine induced NP epitope-specific CTLs, and mice were protected against body weight loss after a challenge with a mouse-adapted A/HK/68 (H3N2) virus. However, a passive serum transfer did not protect mice against heterologous virus replication in the lungs.	[53]
DNA encoding NP of PR8 virus	Mice	i.m.	A total of 50 µg of NP DNA was injected on weeks 0 and 3. Lymphoproliferative responses were detected in splenocytes as early as 2 weeks and as late as 1 year after immunization. Vaccine induced high levels of anti-NP antibodies dominated by IgG2a/b isotypes. Mice were protected against the H3N2 lethal virus, and NP-specific CD4+ and CD8+ T cells both independently acted as effectors.	[54]
DNA encoding NP and M of PR8 virus	Mice	i.m.	A total of 100 μg of two plasmids bearing M and NP genes was administered three times with 2-week intervals. This immunization provided protection against the A/Philippines/2/82/X-79 (H3N2) challenge for intact mice, but not for B6-β2m^(−/−)^ animals defective in CD8+ CTLs. Survival was completely abrogated in the case of simultaneous CD4+ and CD8+ depletion during the challenge.	[87]
DNA and Ad5 vector expressing NP of PR8 virus	Mice	i.m.	Mice were primed with NP-encoding DNA at a dose of 50 μg three times with 2-week intervals, and 2–3 weeks later were boosted with NP-rAd5 at a dose of 10^10^ particles. The higher levels of anti-NP serum IgGs and effector T cells and better protection against the PR8 and A/H5N1 virus challenges were seen in mice immunized by the combined strategy compared to DNA injection alone.	[55]
Ad5 vector expressing consensus NP and M2	Mice	i.n.	Two-dose immunization with 10^7^ and 10^8^ PFU induced high levels of cross-reactive anti-NP and anti-M2 IgGs, which were long-lasting. In addition, antigen-specific IFNγ-secreting CD4+/CD8+ effector T cells and CD8+CD44+ effector memory T cells were induced upon vaccination. Immunized mice were fully protected against a challenge with 10 LD_50_ of the H1N1, H2N3, H3N2, H5N2, or H9N2 human and avian influenza A viruses.	[88]
VSV vector expressing HA or NP of PR8 virus	Mice	i.m. or i.n.	For both administration routes, rVSV-NP was injected once at a dose of 2.5 × 10^6^ PFU, which induced robust NP-specific CD8 T cells with cytotoxic activity in vivo; however, this response was insufficient to protect against a challenge with 100 MLD_50_ of the PR8 virus. A cocktail of rVSV-NP and rVSV-HA (2.5 × 10^6^ PFU each) administered i.n. prevented viral replication in the lungs and weight loss of mice upon a challenge with the same dose of the PR8 virus.	[89]
DNA encoding fusion M2e-NP (NP of swine H3N2 virus)	Pigs	s.c.	Animals immunized with 200 μg of plasmid three times at 3-week intervals appeared to be not protected from a lethal dose (10^8^ TCID_50_) of the A/Sw/Best/96 (H1N1) virus; moreover, vaccinated animals demonstrated more vigorous disease signs that those in the control group.	[90]
Parapox Orf vector expressing HA or HA+NP of swine H1N1 virus	Pigs	i.m.	Immunization of 3-week-old pigs with 10^7^ TCID_50_/mL of OV-HA or OV-HA-NP two times at 3-week intervals revealed a higher magnitude of virus-specific IgG antibody T-cell responses in OV-HA-NP compared to OV-HA. A challenge with a panel of diverse swine influenza viruses revealed the better cross-protective potential of the combined vaccine OV-HA-NP than the OV-HA.	[71]
Ad5 vector expressing NP of B/Yamagata/16/88	Mice	i.m. or i.n.	Vigorous NP-cross-specific humoral and CD8+ T-cell responses were induced after immunization with 1 × 10^8^ FFU using both administration routes, but only intranasal immunization completely protected mice against both lineages of an influenza B virus challenge.	[91]
Ad5 vectors expressing NP of B/Yamagata or B/Victoria virus lineages	Mice	i.n.	Both vaccine candidates were administered once, at a dose of 1 × 10^8^ FFU, which induced high levels of cross-reactive antibodies and CD8+ T cells that recognized the NP epitopes of both influenza B viruses. Vaccination with either candidate led to reduced weight loss and tissue viral loads post-challenge with 5 LD_50_ of both the B/Yamagata and B/Victoria virus lineages.	[16]
MVA vector expressing NP of PR8 virus	Mice	i.m.	Several new chimeric poxviral promoters were assessed in this study. Vaccines were injected once or twice over a 21-day interval at a dose of 10^8^ FFU. All variants were efficient inducers of serum anti-NP antibodies and CD8+ T-cell responses. Two MVA-NP constructs with PVGF and PLMU4 promoters demonstrated improved protection against a lethal dose (10^3^ TCID_50_) of a mouse-adapted PR8 virus, compared to other promoters. No cross-protection against other influenza A virus subtypes was assessed in this study.	[92]
Recombinant NP of PR8 virus	Mice	i.n. or i.p.	The various doses, administration routes, and effects of an adjuvant were tested. The best results—in terms of induction of anti-NP IgGs and IFNγ-producing CD4+ and CD8+ T cells, as well as protection against the PR8, 2009 H1N1, and H9N2 viruses—were demonstrated for 100 µg of full-length rNP combined with C48/80 adjuvant, administered i.n. twice with a 3-week interval.	[72]
Recombinant NP of PR8 virus	Mice	i.n. or i.p.	A total of 100 µg of rNP with cholera toxin B (CTB) adjuvant administered i.n. three times promoted the formation of anti-NP IgGs and IFNγ-producing CD4+ and CD8+ T cells. This scheme conferred complete protection against 10 LD_50_ of the PR8 virus and the heterologous avian H5N1 and H9N2 viruses.	[73]
Recombinant NP of PR8 virus	Mice	i.n.	The rNP was mixed with adjuvant c-di-AMP and i.n. administered at a dose of 10 µg two times with a 3-week interval. The vaccine induced serum and mucosal anti-NP antibodies, promoted the overexpression of INF-γ and IL-2, and reduced lung damage and weight loss after infection with a sublethal dose (0.5 LD_50_) of the PR8 virus challenge on day 60.	[74]
Recombinant NP of A/WSN/1933 (H1N1)	Mice	i.m.	The rNP was fused with the oligomerization domain OVX313, resulting in the formation of a heptamer. This structure was more immunogenic compared to rNP. In particular, 25 µg of the fusion protein injected induced high levels of NP-specific IgGs and CD4+/CD8+ IFNγ; Il-2-, IL-4-, and TNFα-secreting T cells; and lung CD8+ T_RM_ cells. Mice were protected against the challenge with 1–3 LD_50_ of the 2009 H1N1, A/WSN/33, or H3N2 viruses.	[83,84]
Recombinant NP of PR8 virus	Mice	i.m.	A total of 10 µg of rNP adjuvanted with a squalene-based SLA-SE and administered three times with 3-week intervals induced vigorous anti-NP IgG2c antibody responses and NP-specific CD4+ and CD8+ T cells both in young and aged mice. The SLA-SE-adjuvanted rNP vaccination of aged mice resulted in significantly enhanced viral clearance upon a challenge with the PR8 virus.	[75]
Recombinant NP of PR8 virus	Mice	i.n.	rNP fused with the HIV-1 Tat protein or rNP alone was administered three times with 2-week intervals at doses of 10, 30, or 100 µg of TAT-NP. TAT-NP induced higher levels of anti-NP IgGs and IFNγ+ CD4+ and CD8+ T cells than rNP alone, resulting in enhanced protection of the TAT-NP group against 10 LD_50_ of the PR8, H9N2, and H3N2 viruses, compared to rNP.	[78]
Synthetically conserved NP_147–155_ epitope	Mice	i.p.	Mice were immunized with 10 µg of the NPCTL peptide mixed with CpG, MF59, and Alum or Freund’s incomplete adjuvant twice, with a 2-week interval. The NPCTL+CpG adjuvant resulted in a higher production of IFNγ-producing T cells, compared to other formulations, and afforded protection against a lethal challenge with the PR8 virus.	[93]
Recombinant NP and M1 of PR8 virus	Mice	i.d. or i.m.	The effects of coimmunization with the recombinant NP and M1 antigens (two times 21 days apart; 5 µg each) depended on the adjuvant type. The vigorous humoral anti-NP and CD8+ IFNγ+ T-cell responses were observed in the case of the AddaVax adjuvant, whereas only NP+M1 supplemented with RFA could protect mice against a lethal challenge with 4 LD_50_ of the PR8 virus.	[76]
Recombinant NP of A/California/07/2009 (H1N1)	Mice	i.d. or i.m.	The mice were immunized with 5 µg of rNP nonadjuvanted or mixed with CpG 1018 (i.d.) or AddaVax (i.m.) three times at 3-week intervals. The rNP+CpG 1018 regimen induced the highest anti-NP IgG levels and numbers of IFNγ-secreting CD4+ and CD8+ T cells. These mice were completely protected against a challenge with 8 LD_50_ of the mouse-adapted 2009 H1N1 virus.	[77]
Nanoparticles containing NP or HA1 of H9N2	Chickens	i.m.	The chitosan-based nanoparticles contained an inner NP core and an outer HA1 layer. Vaccination with these nanoparticles induced higher titers of anti-HA and anti-NP antibodies and increased the number of activated CD4+/CD8+ T cells, compared with the inactivated vaccine, thus leading to reduced viral shedding upon a challenge.	[80]
Nanoparticles containing NP of A/Aichi/2/1968 (H3N2)	Mice	i.m.	The nanoparticles consisted of an external HA- or M2-enriched layer and an NP-containing core. Being injected two times with a 28-day interval at a dose of 1 µg, the vaccine induced high titers of cross-neutralizing and NP-specific antibodies, as well as NP-specific IFNγ-secreting T cells. The vaccine reduced weight loss, viral replication, and lung tissue damage caused by a lethal infection with 3 LD_50_ of the PR8, H3N2, 2009 H1N1, H5N1, and H7N9 viruses.	[81]
Nanoparticles containing NP and HA of PR8 virus	Mice	i.p.	The NP was encapsulated into a nanocage formed of HA-conjugated apoferritin. Two-dose immunization with 25 µg or nanoparticles elicited both HA- and NP-specific IgG antibodies and protected mice against 10 LD_50_ of the homologous PR8 and heterologous A/FM/1/47 (H1N1) viruses.	[82]
mRNA expressing NP of A/Aichi/2/1968 (H3N2)	Mice and NHP	i.m.	The mice were immunized with lipid nanoparticles containing mRNAs of HA stem and/or NP twice with a 3-week interval at a dose of 5–10 µg, and rhesus macaques were immunized with 300 µg once (if seropositive) or three times (if naïve) with 28-day intervals. The vaccine elicited non-neutralizing cross-reactive antiviral IgGs and NP-specific IFNγ/IL-2-secreting CD4+/CD8+ T cells. Immunization with both HA stem and NP mRNAs induced maximally intensive ADCC responses and better protected mice against lethal doses of distant H1N1 viruses: PR8 and A/California/07/2009.	[18]
mRNA encoding NP of A/Singapore/INFIMH-16-0019/2016 (H3N2)	Mice	i.d.	Mice were immunized with 1–5 µg of lipid-coated mRNAs encoding HA, NA, NP, or M2, twice at a 4-week interval, followed by a challenge with 5 LD_50_ of A/Switzerland/9715293/2013 (H3N2). The vaccines induced high levels of IgGs specific to viral antigens, including neutralizing and CD4/CD8 T-cell responses. The best protective effect was observed in mice immunized with 5 µg of the quadrivalent vaccine.	[52]
ChAdOx1 and MVA vector expressing NP of n A/Panama/2007/1999 (H3N2)	Mice	i.m.	Alternating immunization with 10^8^ infectious units (IU) or PFU of two different vectors bearing both NP and M1 genes two times (3 weeks apart) induced a strong generation of IFNγ-secreting CD8+ T cells, preventing lethal weight loss after infection with 10^3^ of A/Shanghai/1/2013 (H7N9) and provoking more vigorous antiviral IgG responses than vaccination with vectors expressing chimeric HA (cHA) and recombinant HA. Moreover, vaccines expressing HA, M1, and NP simultaneously appeared to better protect the mice from an i.n. challenge with 10^2^ PFU of A/Philippines/2/82 (H3N2) or 10^4^ PFU of A/Jiangxi-Donghu/346/13 (H10N8) than vectors expressing a single antigen.	[66]
ChAdOx1 and MVA vector expressing NP of A/Panama/2007/1999 (H3N2)	Ferrets	i.m.	The prime-boost vaccination (two times with 4-week interval) with 5 × 10^8^ IU (ChAdOx1 prime) and 1 × 10^8^ PFU (MVA boost) of vectors expressing cHA and NP + M1 elicited serum virus-neutralizing antibodies, IgG responses toward the recombinant viral proteins, and generation of IFNγ-producing CD8+ T cells in the mucosa, spleen, and lymph nodes. Vaccine administration also led to decreased viral loads in the upper airways and lungs caused by a challenge with 10^6^ PFU of A/Wyoming/03/2003 (H3N2).	[67]
ChAdOx2 and MVA vector expressing NP of m A/swine/England/1353/2009 (H1N1pdm09)	Pigs	i.m., i.n., aerosol	Pigs pre-exposed i.n. with 3 × 10^6^ PFU of A/swine/England/1353/2009 (H1N1pdm09) were primed with 5 × 10^8^ IU of a ChAdOx2-based vector expressing M1, NA, and NP antigens and boosted 4 weeks after with 1.5 × 10^8^ PFU of MVA-based vector. While aerosol administration route induced high levels of lung cross-specific IgGs and CD4+/CD8+ T cells, i.m. immunization led to strong serum antiviral IgG responses. Both vaccination strategies reduced viral loads and lung lesions upon the challenge with 9 × 10^7^ PFU of A/swine/Ohio/A01354299/2017 (H3N2).	[68]
mRNA expressing NP of A/Michigan/45/2015 (H1N1pdm09)	Ferrets	i.m.	The animals were immunized twice 6 weeks apart with 50 μg of vaccine containing LNP-formulated mRNA of NP, M1, and PB1. The vaccination induced cross-specific IFNγ-secreting CD4+ and CD8+ in peripheral blood and T cells in the respiratory tract and bone marrow. An intratracheal challenge with 10^6^ TCID_50_ A/Anhui/1/2013 (H7N9) revealed reduced body weight loss, pathology score, and viral loads in vaccinated ferrets.	[94]
AAV vector expressing NP of A/Mexico/4603/2009 (H1N1)	Mice	i.m.	AAV-based vaccines (5 × 10^9^ particles) expressing the HA, M1, and NP in combination or alone were administered once. The strong antigen-specific cellular and serum IgG responses were induced in all cases. The maximal protective efficacy, implicating the reduced mortality, morbidity, and viral loads upon the challenge with A/Hamburg//08/2009 (H1N1v), was revealed for the trivalent vaccine, when compared to expressing NP, M1, or HA alone.	[69]
AAV vector expressing NP of A/California/07/2009 (H1N1)	Mice	i.n.	Mice were vaccinated with 1 × 10^11^ viral genomes (vg) of vectors expressing HA, cHA, or NP three times with 3-week intervals. The vaccines induced broadly reactive IgG and IgA antibodies and reduced morbidity and mortality in mice challenged i.n. with the A/California/07/2009 (H1N1) (500 MID_50_) or the PR8 virus (up to 200 MID_50_).	[70]
DNA and Ad5 vectors expressing NP of PR8 and B/Ann Arbor/1/86	Mice	i.m.	The authors compared the efficacy of immunization with cold-adapted (ca) viruses and “DNA encoding NP, M2 or NP+M2 prime—corresponding adenovirus (Ad5) boost” vaccination. Mice were immunized with three doses of DNA (50 µg) 14 days apart and boosted with 10^10^ particles of the Ad5-based vaccines. Boosting doses enhanced the CD4+/CD8+ T-cell memory responses and reduced weight loss after i.n. challenge with 1.5 × 10^4^ LD_50_ of PR8. However, only vaccination with the combined DNA–Ad5 variant reduced morbidity and mortality after the challenge with 10 LD_50_ of the highly pathogenic A/Vietnam/1203/2004 (H5N1) virus.	[56]
DNA and Ad5 vector expressing NP of PR8 and A/Thailand/1(KAN-1)/2004 (H5N1)	Mice, ferrets	i.m.	DNA-based vaccines encoding HA, NP, or M2 proteins were used alone or in combination for priming of animals. Mice were immunized with 15 µg of DNAs three times at 3-week intervals; while ferrets received 250 µg of DNA three times and finally were boosted with 10^10^ particles of the corresponding Ad5-vectored vaccines. The animals were i.n. challenged with 100 LD_50_ (mice) or 10^7^ EID_5_0 (ferrets) of A/Vietnam/1203/2004 (H5N1). All DNA– or DNA–Ad5 vaccinations induced serum IgGs and T-cell responses in a similar way, but only the HA-containing vaccines conferred protection against a lethal H5N1 challenge.	[57]
DNA encoding NP of A/chicken/Qalubiya/1/2006 (H5N1)	Chickens	i.m.	Tandems of expression plasmids encoding NP and each of the seven other antigens of the H5N1 virus were used as DNA vaccines to immunize the chickens. The most immunogenic pairs appeared to be NP/HA and NP/NS.	[95]
DNA or mRNA expressing NP of A/NL/18/94 (H3N2)	Mice	i.m., intranodal	Mice were immunized three times at 2-week intervals, with 100 µg of DNA- or 50 µg of mRNA-expressing vaccine (i.m. or intranodal, respectively). In the latter case, stronger IFNγ+ CTL responses were observed and prolonged when reducing the mRNA dose. In the survival study, mice were prime-boost vaccinated with 17 µg of the mRNA-based vaccine and then infected with 1 LD_50_ of PR8. Immunization mitigated the harmful effects of an infection challenge, lowered T-cell infiltration in the lungs, and increased the levels of IFNγ, IL-2, IL-6, IL-9, IL-10, IL-13, and G-CSF cytokines in the lungs.	[96]
Wyeth strain of vaccinia virus vector expressing NP of A/Vietnam/1203/2004 (H5N1)	Mice	s.c.	A pentavalent vector-based vaccine expressing HA, NA, NP, M1, and M2 antigens and the IL-15 gene injected at a dose of 1 × 10^7^ PFU was able to induce the cross-neutralizing IgG antibodies and robust CD4+/CD8+ IFNγ-secreting T-cell responses in mice. Upon an i.n. challenge with 100 LD_50_ of A/Ck/Indonesia/BL/2003 (H5N1), vaccinated mice demonstrated reduced weight loss and lung damage.	[63]

CTB: cholera toxin B subunit; NHP: nonhuman primate; RFA: radiofrequency adjuvant; PBS: phosphate-buffered saline; CTLs: cytotoxic T lymphocytes; cHA: chimeric HA; Ad5: adenovirus type 5; ChAdOx: chimpanzee adenovirus Oxford; rNP: recombinant nucleoprotein; rVSV: recombinant vesicular stomatitis virus; OV: orf virus; MLD50: mouse median lethal dose; PFU: plaque-forming unit; FFU: focus-forming unit; AMP: adenosine monophosphate; MVA: modified vaccinia virus Ankara; TCID_50_: median tissue culture infectious dose; LD_50_: median lethal dose; IgGs: immunoglobulins class G; IFNγ: interferon gamma; IL-2: interleukin-2; IL-4: interleukin-4; TRM: tissue-resident memory; TNFα: tumor necrosis factor alpha; ADCC: antibody-dependent cellular cytotoxicity; EID_50_: median egg infectious dose; TAT: trans-activator; HIV-1: human immunodeficiency virus 1; CpG: cytosine phosphoguanine; mRNA: matrix ribonucleic acid; MHC: major histocompatibility complex; LNP: lipid nanoparticle; MID: mouse median infective dose.

**Table 2 vaccines-11-01747-t002:** Clinical trials of NP-based influenza vaccines.

Company	Phase	Year	ClinicalTrials.gov Identifier	Available Results
University of Oxford (Oxford, UK)	II	2009–2010	NCT00993083	Yes [60,100]
University of Oxford (Oxford, UK)	I	2008–2012	NCT00942071	Yes [58,59]
University of Oxford (Oxford, UK)	I	2013	NCT01623518	Yes [65]
University of Oxford (Oxford, UK)	I	2013–2015	NCT01818362	Yes [101]
University of Oxford (Oxford, UK)	I	2014	NCT02014168	No
Vaccitech Ltd. (Didcot, UK)	I	2017	NCT03277456	Yes [102]
Vaccitech Ltd. (Didcot, UK)	II	2017–2018	NCT03300362	Yes [61,62]
Vaccitech Ltd. (Didcot, UK)	II	2019	NCT03880474	No
Vaccitech Ltd. (Didcot, UK)	II	2019–2020	NCT03880474	Yes [103]
Vaccitech Ltd. (Didcot, UK)	II	2019–2020	NCT03883113	No
Osivax (Lyon, France)	I	2018–2019	NCT03594890	Yes [85]
Osivax (Lyon, France)	II	2020	NCT04192500	Yes [104]
Osivax (Lyon, France)	II	2021–2022	NCT05060887	No
Osivax (Lyon, France)	II	2022	NCT05284799	No
Osivax (Lyon, France)	II	2022	NCT05060887	Yes [86].
Osivax (Lyon, France)	II	2023–2024	NCT05569239	No
Osivax (Lyon, France)	II	2023–2024	NCT05734040	No
BiondVax Pharmaceuticals Ltd. (Jerusalem, Israel)	I/II	2009	NCT00877448	Yes [97]
BiondVax Pharmaceuticals Ltd. (Ltd. (Jerusalem, Israel)	I/II	2009–2010	NCT01010737	No
BiondVax Pharmaceuticals Ltd. (Ltd. (Jerusalem, Israel)	II	2010–2011	NCT01146119	No
BiondVax Pharmaceuticals Ltd. (Ltd. (Jerusalem, Israel)	II	2011–2012	NCT01419925	Yes [99]
BiondVax Pharmaceuticals Ltd. (Ltd. (Jerusalem, Israel)	II	2014–2015	NCT02293317	No
BiondVax Pharmaceuticals Ltd. (Ltd. (Jerusalem, Israel)	III	2018–2020	NCT03450915	No
NIAID (Bethesda, USA)	II	2018–2019	NCT03058692	Yes [105]
PepTcell Ltd. (London, UK)	I	2010	NCT01181336	Yes [106,107]
PepTcell Ltd. (London, UK)	I	2010	NCT01226758	Yes [108]
PepTcell Ltd. (London, UK)	II	2016–2017	NCT03180801	Yes [109]
PepTcell Ltd. (London, UK)	II	2016–2017	NCT02962908	Yes [110,111]
Immune Targeting Systems Ltd. (London, UK)	I	2010–2011	NCT01265914	Yes [112]
Immune Targeting Systems Ltd. (London, UK)	I	2012	NCT01677676	No
Immune Targeting Systems Ltd. (London, UK)	I	2012–2013	NCT01701752	No
Immune Targeting Systems Ltd. (London, UK)	I/II	2014	NCT02071329	No

## Data Availability

Data sharing not applicable.

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
