# Peer review of "Nucleoprotein as a Promising Antigen for Broadly Protective Influenza Vaccines"

_vaccines, 2023, doi:10.3390/vaccines11121747_

Round 1

Reviewer 1 Report

Comments and Suggestions for Authors

The authors reviewed the current knowledge of influenza A viral nucleoprotein, the development of experimental NP-based vaccine prototypes together with the data of preclinical experiments with a focus on cross-reactive T-cell responses, analysis of variability of NP sequences and approaches to improving the efficacy of influenza virus vaccines. The manuscript is considered informative, however, the following issues need to be addressed.

 It seems necessary to summarise mechanism of a broad immune response induced by NP.

 The data from a combination of NP with HA or NA vaccines authors may be explored further.

 The implication of the manuscript in Concluding remarks appeared to be highlighted.

 As mentioned in Line 339, numerous critical amino acid substitutions which occurred in NP protein need to be summarized.  

 Regarding the scientific writing, the whole manuscript seems to be a bit concise. For examples, Table 1: Main results would be recommended in the description of phrases rather than the sentences. Keep some abbreviations e.g.  i.n. and  i.m. consistent in the whole manuscript. The abbreviations e.g. ADCC, TAT should have their full names when you use it for the first time. Some grammar errors across the whole manuscript, E.g. Line 150 (For this), line 196 (resulted- resultant).  

Comments on the Quality of English Language

The manuscript needs to be revised towards the publication. 

Author Response

  1. It seems necessary to summarise mechanism of a broad immune response induced by NP.

Authors’ response: We thank the reviewer for this note. The main immune responses triggered by NP are mentioned in the Introduction and Section 3. However, we additionally supplemented the Section 2 with the paragraph devoted to the defense mechanisms stimulated by NP.

  1. The data from a combination of NP with HA or NA vaccines authors may be explored further.

Authors’ response: We thank the reviewer for this suggestion. We provided an additional information on the combined vaccine candidates containing NP and HA/NA antigens in the corresponding paragraph.

  1. The implication of the manuscript in Concluding remarks appeared to be highlighted.

Authors’ response: We modified the Concluding remarks paragraphs to emphasize the main message of the manuscript.

  1. As mentioned in Line 339, numerous critical amino acid substitutions which occurred in NP protein need to be summarized.

Authors’ response: The data on the point mutations which were shown to affect the CTL epitopes were added to the corresponding paragraph of Section 4.

  1. Regarding the scientific writing, the whole manuscript seems to be a bit concise. For examples, Table 1: Main results would be recommended in the description of phrases rather than the sentences. Keep some abbreviations e.g.  i.n. and  i.m. consistent in the whole manuscript. The abbreviations e.g. ADCC, TAT should have their full names when you use it for the first time. Some grammar errors across the whole manuscript, E.g. Line 150 (For this), line 196 (resulted- resultant).

Authors’ response: We thank the reviewer for this suggestion. We believe that writing the manuscript concisely allows making the phrases more meaningful. We made the commonly used abbreviations consistent throughout the manuscript, have deciphered the abbreviations that appear in the text for the first time and corrected the mentioned grammar errors. The manuscript was also subjected by a professional English language editing by the MDPI’s services.

Reviewer 2 Report

Comments and Suggestions for Authors

The manuscript " Nucleoprotein as a Promising Antigen for Broadly Protective Influenza Vaccines" was a good summary of NP-based influenza vaccine. However, the presentation of this manuscript is not appropriate and should be revised thoroughly.

1) This is a review-type paper. However, section 4 incorporated a large amount of alignment results,  figures, and tables generated by authors. These data should not be included in the review paper. Authors may describe what they found from a further digging of published studies in brief to suggest new strategies. However, it is not appropriate to include a large amount of unpublished tables and figures in the review manuscript.

2) The organization/structure of the manuscript needs to be adjusted. I would suggest:

a.  Review the past studies on NP-based vaccines for the pros and cons. 

b. Review the pre-clinical and clinical trials of NP-targeted influenza vaccine for the success and failure in the past.

c. Point out the possible impact due to the variability of NP, which you can briefly describe what you found from alignment and immune epitope conservancy analysis (not tables/figures, please).

d. Suggest new strategies for vaccine design, such as conservancy analysis to choose the epitopes, optimize genome composition, optimize adjuvant and delivery vehicles et al.

The authors did an excellent job on the knowledge preparation for this review. I'm looking forward to reading the updated version of the manuscripts.

Comments on the Quality of English Language

A few examples of confusing sentences. Authors may want to do fresh-eyes proofreading.

1. Line 120-line 122. 

2. Line 136-138. You may want to mention the result of the control group as a comparison.

3. Line 174- 200, vector platforms can be listed in a table for easier presentation and comparison.  Same suggestion for the adjuvant summary part as well

4. Line 485-491. used "should", my question is whether these changes really achieved the result claimed or just guessing.

Author Response

1) This is a review-type paper. However, section 4 incorporated a large amount of alignment results, figures, and tables generated by authors. These data should not be included in the review paper. Authors may describe what they found from a further digging of published studies in brief to suggest new strategies. However, it is not appropriate to include a large amount of unpublished tables and figures in the review manuscript.

Authors’ response: We agree with the reviewer that the review paper should have only references to the published data. However, it was critically important to demonstrate to the readers that the NP sequences of older viruses do differ from the ones of currently circulating viruses. Therefore, we removed most of our own data, but left one NP alignment figure in the main text, and the other two alignments were moved to Supplementary materials.

2) The organization/structure of the manuscript needs to be adjusted. I would suggest:

  1. Review the past studies on NP-based vaccines for the pros and cons. 
  2. Review the pre-clinical and clinical trials of NP-targeted influenza vaccine for the success and failure in the past.
  3. Point out the possible impact due to the variability of NP, which you can briefly describe what you found from alignment and immune epitope conservancy analysis (not tables/figures, please).
  4. Suggest new strategies for vaccine design, such as conservancy analysis to choose the epitopes, optimize genome composition, optimize adjuvant and delivery vehicles et al.

Authors’ response: We thank the reviewer for these suggestions. We believe that the structure of our review generally follows the organization proposed by the reviewer. We added additional thoughts on the new strategies that can be utilized to improve the performance of NP-based vaccines to the Section 5.

Reviewer 3 Report

Comments and Suggestions for Authors

Comments for the authors of Vaccines manuscript vaccines-2679436:

The author of the Vaccines manuscript “Nucleoprotein as a Promising Antigen for Broadly Protective Influenza Vaccines”, review the potential use of influenza virus nucleoprotein as a candidate for a broadly protective influenza vaccine.  The overall issue is the variability of the target antigens for influenza, hemagglutinin (HA) and neuraminidase (NA).  Since the nucleoprotein (NP) is more conserved it seems to be a more likely target for a broadly protective vaccine.  The NP is not represented in subunit or recombinant protein influenza vaccines, but it is expressed after inoculation with live, attenuated influenza virus vaccines.  The immunity induced against NP includes both antibodies and T cells.  The authors note the low-level changes in NP that can affect vaccine efficacy and immunogenicity, and summarize the pre-clinical and clinical trials on vaccines that induce NP-specific immunity.  Below are some comments that the authors should consider while revising their work.   

General Comments:

  1. Throughout the text, there are multiple times when the sentence and paragraph structure could be improved.  One notable section is section 2: Nucleoprotein structure and function.  Please review and revise as appropriate.
  2. The sentence on line 83-84 has the number 24 included in a way that is not completely clear.  Please clarify the use of 24 in that sentence.
  3. The review does an excellent job of summarizing the current knowledge of anti-NP immunity in both the laboratory and the clinical setting. 
  4. Table 1 is especially helpful for seeing what has been done in mice, pigs, chickens, and NHP.  Table 2 presents human studies and whether results are available or not.
Comments on the Quality of English Language
  1. Throughout the text, there are multiple times when the sentence and paragraph structure could be improved.  One notable section is section 2: Nucleoprotein structure and function.  Please review and revise as appropriate.

Author Response

  1. Throughout the text, there are multiple times when the sentence and paragraph structure could be improved.  One notable section is section 2: Nucleoprotein structure and function.  Please review and revise as appropriate.

Authors’ response: We thank the reviewer for this note. The text structure of Section 2 was reviewed and improved. And the English language usage was corrected by professional editors.

  1. The sentence on line 83-84 has the number 24 included in a way that is not completely clear.  Please clarify the use of 24 in that sentence.

Authors’ response: The number 24 indicates the number of nucleotides to which the NP molecule of influenza virus binds. This piece of text has been clarified.

  1. The review does an excellent job of summarizing the current knowledge of anti-NP immunity in both the laboratory and the clinical setting. 

Authors’ response: We thank the reviewer for this positive feedback.

  1. Table 1 is especially helpful for seeing what has been done in mice, pigs, chickens, and NHP.  Table 2 presents human studies and whether results are available or not.

Authors’ response: We thank the reviewer for appreciating the value of our work.

Round 2

Reviewer 1 Report

Comments and Suggestions for Authors

The authors made great efforts to address the reviewer's comments and suggestions towards the publication. The manuscript may need further language check before publication. 

Comments on the Quality of English Language

Some grammatical erors exist in the manuscrit e.g.  Line 39: such as ...AND  and so on. cannot be used in one sentence.  Line 365: "not commervially available" seems better.  

Author Response

The authors made great efforts to address the reviewer's comments and suggestions towards the publication. The manuscript may need further language check before publication. 

Authors' response: we thank the reviewer for the positive feedback. Additional English language usage corrections were made using specified services.

Reviewer 2 Report

Comments and Suggestions for Authors

Great work on revision. Very impressive! It is a comprehensive review of NP-based vaccines.

Author Response

Great work on revision. Very impressive! It is a comprehensive review of NP-based vaccines.

Authors' response: we thank the reviewer for this positive feedback.